# Reassessing the Role of Potassium in Tomato Grown with Water Shortages

Anna De Luca [1,2], Mireia Corell [3], Mathilde Chivet [4], M. Angeles Parrado [1], José M. Pardo [2] and Eduardo O. Leidi [1,*]

1   Department of Plant Biotechnology, IRNAS-CSIC, Avenida Reina Mercedes 10, 41012 Sevilla, Spain; anna.luca@ibvf.csic.es (A.D.L.); maparrado@irnase.csic.es (M.A.P.)
2   IBVF-CSIC, Avenida Americo Vespucio 49, 41092 Sevilla, Spain; jose.pardo@csic.es
3   Departamento de Ciencias Agroforestales (ETSIA), Universidad de Sevilla, Ctra. Utrera Km. 1, 41013 Sevilla, Spain; mcorell@us.es
4   Grenoble Institut Neurosciences (GIN), Université Grenoble Alpes, Inserm, U1216, 38000 Grenoble, France; mathilde.chivet@gmail.com
*   Correspondence: eo.leidi@csic.es

**Abstract:** Potassium (K) is closely related to plant water uptake and use and affects key processes in assimilation and growth. The aim of this work was to find out to what extent K supply and enhanced compartmentation might improve water use and productivity when tomato plants suffered from periods of water stress. Yield, water traits, gas exchange, photosynthetic rate and biomass partition were determined. When plants suffered dehydration, increasing K supply was associated with reduction in stomatal conductance and increased water contents, but failed to protect photosynthetic rate. Potassium supplements increased shoot growth, fruit setting and yield under water stress. However, increasing the K supply could not counteract the great yield reduction under drought. A transgenic tomato line with enhanced K uptake into vacuoles and able to reach higher plant K contents, still showed poor yield performance under water stress and had lower K use efficiency than the control plants. With unlimited water supply (hydroponics), plants grown in low-K showed greater root hydraulic conductivity than at higher K availability and stomatal conductance was not associated with leaf K concentration. In conclusion, increasing K supply and tissue content improved some physiological features related to drought tolerance but did not overcome yield restrictions imposed by water stress.

**Keywords:** *Solanum lycopersicum*; drought potassium; vacuolar transporter

## 1. Introduction

Potassium (K) is the most abundant cation and plays important roles in plant growth and development where it contributes to charge balance, osmotic adjustment and enzyme catalysis [1]. Stored in vacuoles, K contributes to generating the osmotic pressure required for turgor and expansive growth. Potassium contributes to water uptake and the generation of turgor pressure in the stomatal guard cells and the phloem [1]. At low K availability, plants are more susceptible to wilting in drying soils and a mild K deficiency affects photosynthesis by reducing stomatal conductance and photosynthetic biochemical reactions [2]. High K availability might counteract the inhibitory effect of drought on $CO_2$ assimilation by protecting photosynthetic capacity [3,4].

Plant K uptake is affected by available water as low soil water content reduces mass flow and diffusion of the nutrients to the rhizosphere, and simultaneously decreases root growth [5]. Potassium transport into aerial parts does not rely only on leaf transpiration, and other factors such as root pressure contribute to long distance transport [6]. However, greater water flow by active transpiration increases K flux to the shoot and K uptake by the roots when it is available [7]. In doing so, symplastic K transport depends on

membrane transport proteins such as high-affinity KUP/HAK transporters and K-selective channels, which facilitate cation movement between plant cells and its final delivery into the xylem [8]. During drought, K fluxes into shoots become reduced by less transpired water (lower xylem water flow) and reduced expression of K-loading channels into the xylem [7].

Tomato requires high K supplements for high yields and optimum fruit quality [9] and low K affects flower development and fruit setting efficiency [10]. When K is scarce, tomato plants reduce sink activity (stems and fruits) prior to having any effect on photosynthesis [11,12]. Considering the great importance of K supply and water availability for fruit yield and quality, we aimed at studying how K availability and compartmentation into vacuoles might affect plant water use and productivity under water stress. The overexpression of a vacuolar K,Na/H antiporter (NHX1) contributes to salt-tolerance in tomato by increasing root K uptake and its storage plant vacuoles [8,13]. By including a transgenic line able to take up more of this nutrient, we tried to assess to what extent a greater plant K content might affect some physiological processes under stress. Specifically, the questions addressed were whether a greater K vacuolar accumulation might contribute to plant water acquisition and use for increasing productivity, and also if enhanced K capture might reduce water loss and provide greater protection of photosynthetic activity, resulting in increased water-use efficiency.

## 2. Materials and Methods

### 2.1. Plant Material and Stress Treatments

Two tomato lines, the cultivar MicroTom (henceforth, the wild-type, WT) [14] and a transgenic line (N367), were used. The line N367 overexpresses the vacuolar AtNHX1 cation/proton antiporter and is able to accumulate greater K quantities in vacuoles and tissues [13]. Seeds were pregerminated (28 °C, 4 days) and the seedlings were transplanted into quartz sand (0.5 L pots) and irrigated every three days with a modified Hewitt's nutrient solution supplemented with K at the concentrations required for each experiment. The basic nutrient solution contained (in mM): $NO_3^-$, 4; $H_2PO_4^-$, 1; $SO_4^{2-}$, 3; $Ca^{2+}$, 2; $Mg^{2+}$, 1; and (in µM): Fe, 50 (as FeEDDHA); Mn, 10; Cu, 1; Zn, 5; B, 30. Three different K concentrations were used as nutritional treatments by supplementing the basic solution with 0.1, 1 and 10 mM K (as $K_2SO_4$). When 0.1 mM K was added, 1 mM phosphate was provided as $NaH_2PO_4$. For addition of 10 mM K, Mg concentration in the solution was increased to 5 mM to avoid Mg deficiency [9]. Plants were grown in a greenhouse (mean temperature, 26 °C; mean relative humidity, 45.0%).

When plants reached the first flowering stage, water stress treatments were initiated by withholding irrigation during 2 weeks followed by one day with abundant watering (to flush pot sand) with nutrient solution supplemented with K as required. This treatment was repeated several times, depending on the experiment (Supplementary Materials Figure S1). When measuring yield, fruits were collected after six cycles of drying/watering, while control plants kept a regular watering regime every 3 days. To test the effect of acclimation, plants were treated with either one or four cycles of watering/drying. Plants submitted to a single water stress cycle were considered "nonacclimated", whereas plants that experienced four consecutive watering/drying cycles were judged as "acclimated". This preconditioning (priming) to stress allows osmotic adjustment and expression of drought-tolerance mechanisms [15,16]. Physiological measurements (leaf water status, photosynthesis, stomatal conductance) were taken at the end of the drying periods and before rewatering, as stated in the Results section. Control plants were irrigated normally every three days.

### 2.2. Leaf Water Status

Leaf water status was estimated by measuring relative water content (RWC) and leaf water content (LWCT) [17]. Leaf osmotic potentials were measured in leaf samples by using

a thermocouple psychrometer (HR33 microvoltmeter and C-52 sample chamber, Wescor, Logan, UT, USA).

### 2.3. Gas Exchange and Photochemical Reflectance Index

Gas exchange measurements were performed in the youngest fully expanded leaves with an LC-Pro Photosynthesis System gas analyzer (ADC Bioscientific, Hoddesdon, UK). Leaf water-use efficiency (WUEi) was estimated as the ratio between photosynthesis (A) and stomatal conductance (gs). The photochemical reflectance index (PRI) was determined with a PlantPen PRI 200 (PSI, Drásov, Czech Republic).

### 2.4. Hydroponic Culture and Root Hydraulic Conductivity

For determining the effect of K supply on root hydraulic conductivity, a critical component of plant water uptake and transport, 1-week old seedlings of both MicroTom lines, WT and N367, were transferred into aerated nutrient solutions contained in 10-L plastic boxes as previously described [13] and cultivated hydroponically until flowering in the greenhouse.

Root hydraulic conductivity ($L_0$) was estimated by measuring root exudation rates in detopped plants [18] grown hydroponically with nutrient solutions containing different K concentrations [13]. Xylem saps exuded from roots during 1 h were collected in Eppendorf tubes with micropipettes and frozen at $-20\ ^\circ C$ for further analysis of K concentration.

### 2.5. Analysis of K Concentration

Potassium was determined in leaves used for porometry and gas exchange, as well as in remaining leaves, stems, roots and fruits, or in xylem saps from root hydraulic conductivity measurements. Plant material (leaves, stems, roots and fruits) was weighed before and after drying (48 h, $70\ ^\circ C$), and ground to powder. Potassium was extracted with distilled water by heating ($90\ ^\circ C$, 1 h) and sonication, and measured by flame photometry (PFP7, Jenway, Staffordshire, UK). Potassium in xylem saps was measured with the same equipment after appropriate dilutions.

### 2.6. Harvest

Fruits were harvested after plants had received six watering/drying cycles starting immediately after the onset of flowering. At harvest, shoot, fruit and root fresh and dry weights and fruit number were recorded. Sugar contents in dried leaves, stems and roots were estimated after hot water extraction with the phenol–sulphuric method using a sucrose standard [19]. Soluble solids in fruits were estimated by refractometry as Brix degrees (Atago Hand held refractometer). Harvest index was calculated as the ratio among fruit weight and total plant weight. Potassium use efficiency (KUE) was calculated by dividing the fruit mass by the total plant K uptake.

### 2.7. Free Amino Acid Contents in Leaves and Flowers

Considering the effect of available K on plant amino acids concentration [20,21] and the protection provided by compatible solutes against drought stress [22], we measured free amino acids in leaves and flowers of water-stressed tomato under two K regimes. At full flowering stage, and after two cycles of watering/drying, fully expanded leaves and flowers from plants grown at 0.1 and 10 mM K were frozen in liquid $N_2$ and stored at $-70\ ^\circ C$. Then, they were homogenized in Eppendorf tubes with plastic pestles and centrifuged (10,000 rpm, 5 min at $4\ ^\circ C$) and free amino acids in the supernatants were separated and quantified after derivatization with phenylisotiocyanate by reversed-phase high-performance liquid chromatography [23].

### 2.8. Statistical Analysis

The experimental design was a complete randomized design with a factorial $3 \times 2$ (3 K concentrations, 2 water treatments) with 3–4 plants per treatment or a factorial $3 \times 2$

(3 K concentrations, 2 tomato lines) with four plants per treatment. Analysis of variance (ANOVA) was applied in order to examine significant differences between treatments (K levels, watering) among variables (plant growth parameters, gas exchange measurements, etc.). Normality of the data was checked by Shapiro–Wilks test. When differences were statistically significant, a post hoc test (Least Significant Difference, LSD) was applied for comparisons. A correlation analysis (Pearson) was done between the K concentration in leaves and gas exchange or PRI values, and between plant sugar and water content. Statistical analyses were performed with a standard package (Statistica, StatSoft Inc., Tulsa, OK, USA).

## 3. Results

To test the combined effect of K supply and water stress on the vegetative growth and yield, wild-type MicroTom plants were grown in sand and irrigated every 3 days with nutrient solution containing 0.1, 1 or 10 mM K. When plants reached the flowering stage, plants of each group were either irrigated normally or submitted to six consecutive cycles of drying and rewatering with nutrient solution (Supplementary Figure S1). Then, plants and fruits were harvested, and growth and yield parameters were measured. Results showed that water stress and the concentration of supplied K greatly affected fruit setting and therefore fruit yield (Table 1).

**Table 1.** Effect of K supply and watering treatment on plant growth and fruit production in tomato. Control plants (irrigated) were compared to plants submitted to six cycles of drying/watering after flower initiation (drought). Means followed by the same letter are not significantly different (LSD Test, *p* < 0.05). K and W in ANOVA represent potassium and water stress treatment factors, respectively.

| K Supply (mM) | Water Treatment | Shoot (g) | Root (g) | Shoot/ Root Ratio | Number of Fruits | Fruit Size (g/fruit) | Yield (g/plant) |
|---|---|---|---|---|---|---|---|
| 0.1 | drought | 4.4 ± 0.9 a | 1.6 ± 0.4 a | 2.7 ± 0.5 a | 0 | 0 | 0 |
| | irrigated | 23.6 ± 3.4 b | 3.4 ± 0.8 b | 7.2 ± 1.4 b | 7 ± 4.5 b | 1.9 ± 1.2 ab | 9.0 ± 3.5 b |
| 1 | drought | 7.9 ± 1.1 c | 1.9 ± 0.7 a | 4.5 ± 1.4 c | 2 ± 1.2 a | 1.3 ± 0.8 b | 2.0 ± 1.5 ac |
| | irrigated | 34.2 ± 5.1 d | 4.5 ± 1.3 c | 7.9 ± 1.5 b | 11 ± 1.6 c | 1.9 ± 0.5 ab | 20.4 ± 3.2 d |
| 10 | drought | 9.6 ± 1.8 c | 2.2 ± 0.9 a | 4.8 ± 1.7 c | 2 ± 1.7 a | 1.7 ± 0.4 ab | 3.3 ± 2.6 c |
| | irrigated | 29.0 ± 3.2 e | 4.1 ± 0.6 c | 7.1 ± 0.9 b | 9 ± 2.8 d | 2.1 ± 0.5 a | 17.2 ± 2.7 e |
| ANOVA | | | | | | | |
| K | | *p* < 0.001 | *p* < 0.01 | *p* < 0.01 | *p* < 0.001 | *p* < 0.001 | *p* < 0.001 |
| W | | *p* < 0.001 | *p* < 0.001 | *p* < 0.001 | *p* < 0.001 | *p* < 0.001 | *p* < 0.001 |
| K*W | | *p* < 0.001 | ns | *p* < 0.05 | ns | *p* < 0.01 | *p* < 0.01 |

Water stress after flowering significantly affected fruit yield but a greater K supply could partially counteract its negative impact on yield (Table 1). Plants growing in 0.1 K and submitted to water stress failed to produce fruits, and this could be prevented partially by increasing the K availability. However, when irrigation was restricted, the inhibition on fruit set had a much greater effect than K limitation. The same was true even when including line N367 (Table 2), which paradoxically showed a greater drought-induced reduction in fruit set and yield in comparison with the control line (WT) in spite of the higher K contents in the former (Table 3). Under regular irrigation, low K significantly reduced fruit set, and therefore, fruit yield. A surplus of K improved plant performance under drought by increasing shoot mass (greater photosynthetic area) while root mass was unaffected (Table 1).

**Table 2.** Effects of K supply on fruit yield, solute contents (Brix degrees) and fruit K concentration in two lines (wild-type (WT) and N367) differing in K uptake after six cycles of limited watering (see Section 2). Means followed by the same letter are not significantly different (LSD Test, $p < 0.05$). K and L in ANOVA represent potassium and line factors, respectively.

| K Supply (mM) | Line | Fruit Number | Fruit Size (g/fruit) | Fruit Yield (g/plant) | Brix (°) | K (%) |
|---|---|---|---|---|---|---|
| 0.1 | WT | 3 ± 1.1 ab | 0.8 ± 0.7 a | 1.8 ± 1.3 a | 5.1 ± 0.4 a | 21.6 ± 3.5 a |
| | N367 | 1 ± 0.6 a | 0.2 ± 0.1 a | 0.2 ± 0.1 a | 5.9 ± 0.1 ab | 27.9 ± 2.2 a |
| 1 | WT | 5 ± 2.6 ab | 2.5 ± 0.9 c | 10.5 ± 2.2 b | 5.9 ± 0.8 ab | 32.3 ± 5.3 ab |
| | N367 | 6 ± 2.1 bc | 1.0 ± 0.5 ab | 6.0 ± 1.3 c | 6.7 ± 0.8 b | 43.0 ± 14.7 b |
| 10 | WT | 7 ± 4.8 bc | 1.9 ± 0.6 bc | 11.0 ± 2.8 b | 6.6 ± 0.9 b | 60.5 ± 4.3 c |
| | N367 | 10 ± 2.8 c | 0.9 ± 0.4 a | 9.0 ± 3.1 bc | 6.2 ± 0.9 b | 60.6 ± 9.2 c |
| ANOVA | | | | | | |
| K | | $p < 0.01$ | $p < 0.01$ | $p < 0.001$ | $p < 0.05$ | $p < 0.001$ |
| L | | ns | $p < 0.01$ | $p < 0.01$ | ns | ns |
| K*L | | ns | ns | ns | ns | ns |

**Table 3.** Effects of K supply on plant growth and K concentration in different organs of two tomato lines (WT and N367) at harvest after receiving six cycles of limited watering. Means followed by the same letter are not significantly different (LSD Test, $p < 0.05$). K and L in ANOVA represent potassium and line factors, respectively. FW, fresh weight.

| K Supply (mM) | Line | Growth (g FW/plant) | | | K Concentration (%) | | |
|---|---|---|---|---|---|---|---|
| | | Leaves | Stems | Roots | Leaves | Stems | Roots |
| 0.1 | WT | 2.68 ± 0.32 a | 2.20 ± 0.45 a | 2.10 ± 0.35 ab | 0.28 ± 0.10 a | 0.38 ± 0.11 a | 0.13 ± 0.07 a |
| | N367 | 2.30 ± 0.21 a | 3.50 ± 0.28 b | 1.48 ± 0.19 a | 0.68 ± 0.15 ab | 1.10 ± 0.10 a | 0.28 ± 0.08 a |
| 1 | WT | 3.48 ± 0.95 ab | 2.85 ± 0.77 ab | 2.86 ± 1.34 b | 1.97 ± 0.51 b | 2.70 ± 0.43 b | 0.25 ± 0.02 a |
| | N367 | 2.85 ± 1.20 a | 3.40 ± 0.64 b | 1.80 ± 0.60 a | 3.56 ± 0.46 c | 4.29 ± 0.85 c | 0.93 ± 0.27 a |
| 10 | WT | 4.68 ± 1.64 b | 3.68 ± 1.28 b | 2.23 ± 0.67 ab | 5.20 ± 1.57 d | 5.95 ± 0.49 d | 3.43 ± 0.49 b |
| | N367 | 3.70 ± 0.57 ab | 3.47 ± 0.09 b | 1.48 ± 0.14 a | 4.44 ± 1.65 cd | 8.00 ± 1.52 e | 5.41 ± 1.80 c |
| ANOVA | | | | | | | |
| K | | $p < 0.001$ | ns | ns | $p < 0.001$ | $p < 0.001$ | $p < 0.001$ |
| L | | ns | ns | $p < 0.01$ | ns | $p < 0.01$ | $p < 0.01$ |
| K*L | | ns | ns | ns | ns | ns | Ns |

The NHX1-overexpressing line N367 failed to improve the growth of leaves, stems or roots at harvest when compared with the control line (WT) in spite of accumulating more K than the WT (Table 3). A significant effect of K on fruit yield was found when K was increased from 0.1 to 1 mM K whereas a further increase in K supply had a marginal effect (Table 2). Surprisingly, N367 with the highest plant K contents (Table 3) showed a tendency towards producing more fruits than the WT as the K availability increased, but had smaller fruits and lower yield than the WT (Table 2). The ANOVA showed a "line" factor (WT vs N367) only significant for fruit size and yield, whereas the "K" factor was critical in all fruit characters measured (Table 2).

Increasing K supply significantly favored fruit K and soluble solid (Brix) accumulation in both lines at harvest (Table 2). While no differences in harvest index were found between lines, a significant lower K use efficiency was recorded for line N367 (Table 4).

**Table 4.** Differences in harvest index and K use efficiency between two tomato lines (WT and N367) grown with different K availabilities and subjected to six cycles of drying/watering. In columns, means followed by the same letter are not significantly different (LSD Test, $p < 0.05$). K and L in ANOVA represent potassium and line factors, respectively.

| K Supply (mM) | Line | Harvest Index | K Use Efficiency |
|---|---|---|---|
| 0.1 | WT | 1.4 a | 13.1 a |
| | N367 | 1.0 a | 3.4 b |
| 1 | WT | 1.3 a | 2.2 bc |
| | N367 | 0.8 a | 0.7 c |
| 10 | WT | 1.2 a | 0.8 bc |
| | N367 | 1.1 a | 0.5 c |
| | ANOVA | | |
| | K | ns | $p < 0.001$ |
| | L | $p < 0.1$ | $p < 0.001$ |
| | K*L | ns | $p < 0.001$ |

Next, we determined key physiological parameters in wild-type and N367 grown with 1 mM K in the nutrient solution until flowering (to avoid severe effects of the lowest K supply on growth) and then submitted to different regimes of K availability (0.1, 1 and 10 mM K). At flowering, both lines were submitted as in the previous experiment to six rounds of 2-week drought and one day of rewatering with the corresponding nutrient solution. In drought-treated wild-type plants, shoot and root water contents were improved by a greater supply of K (Table 5). The same trend was found for lines WT and N367, which at harvest showed greater water content in leaves (0.1 vs. 1–10 mM K) or in roots (0.1–1 vs. 10 mM K) (Table 6). However, stem water contents remained similar in spite of significant increases in their K concentration by increasing K supply (see Table 6).

**Table 5.** Effect of K supply and watering on shoot and root water content (g $H_2O$ g dry weight$^{-1}$) in tomato at harvest. Control plants (irrigated) were compared to plants submitted to six cycles of drying/watering after flower initiation (drought). Means followed by the same letter are not significantly different (LSD Test, $p < 0.05$). K and W in ANOVA represent potassium and water factors, respectively.

| K Supply (mM) | Water Treatment | Shoot Water Content | Root Water Content |
|---|---|---|---|
| 0.1 | drought | $5.1 \pm 0.7$ a | $3.0 \pm 1.0$ a |
| | irrigated | $7.3 \pm 0.6$ b | $6.3 \pm 0.9$ bc |
| 1 | drought | $6.1 \pm 0.5$ c | $4.8 \pm 1.0$ d |
| | irrigated | $7.7 \pm 0.4$ b | $6.9 \pm 1.0$ be |
| 10 | drought | $6.6 \pm 0.6$ c | $5.8 \pm 0.8$ b |
| | irrigated | $7.5 \pm 0.6$ b | $7.2 \pm 0.8$ e |
| | ANOVA | | |
| | K | $p < 0.001$ | $p < 0.01$ |
| | W | $p < 0.001$ | $p < 0.001$ |
| | K*W | $p < 0.001$ | $p < 0.01$ |

Plants receiving less K showed clear wilting symptoms at the end of each drought cycle in comparison with plants receiving a higher K supply (Figure 1).

At harvest, sugar accumulation in leaves was recorded in drought-treated WT and N367 tomato plants at the lowest K supply (Table 6) but no association was found between sugar accumulation and water content, with the exception of roots, which showed a significant correlation between their sugar and water contents (r = 0.61, $p < 0.001$, n = 24). The K concentration provided also had a significant effect on the water content of leaves, stems and roots (Table 6).

**Table 6.** Water and sugar contents in leaves, stems and roots of tomato WT and N367 lines at harvest when grown with different K concentrations and subjected to six cyclical drought periods. Means followed by the same letter are not significantly different (LSD Test, $p < 0.05$). K and L in ANOVA represent potassium and line factors, respectively.

| K Supply (mM) | | Water Content (g g dry wt$^{-1}$) | | | Sugars (%) | | |
| --- | --- | --- | --- | --- | --- | --- | --- |
| | Line | Leaves | Stems | Roots | Leaves | Stems | Roots |
| 0.1 | WT | $4.5 \pm 2.0$ a | $7.4 \pm 1.3$ a | $5.9 \pm 1.5$ a | $16.4 \pm 4.7$ ab | $6.8 \pm 1.1$ a | $3.6 \pm 1.3$ a |
| | N367 | $4.4 \pm 1.3$ a | $6.4 \pm 1.5$ a | $5.6 \pm 1.4$ a | $20.4 \pm 4.9$ a | $11.8 \pm 2.3$ bc | $4.9 \pm 1.5$ a |
| 1 | WT | $6.3 \pm 0.3$ b | $5.9 \pm 0.3$ a | $4.7 \pm 1.5$ a | $12.5 \pm 1.4$ bc | $13.1 \pm 3.5$ bd | $4.7 \pm 3.0$ a |
| | N367 | $6.1 \pm 1.5$ b | $6.1 \pm 0.3$ a | $5.9 \pm 0.7$ a | $13.1 \pm 1.2$ bc | $10.6 \pm 3.9$ abc | $4.6 \pm 1.2$ a |
| 10 | WT | $7.2 \pm 0.5$ b | $6.1 \pm 0.6$ a | $9.2 \pm 1.5$ b | $13.9 \pm 3.5$ bc | $16.0 \pm 3.2$ d | $11.9 \pm 3.3$ b |
| | N367 | $6.9 \pm 0.5$ b | $7.1 \pm 0.4$ a | $10.9 \pm 2.0$ b | $8.9 \pm 2.5$ c | $8.3 \pm 1.2$ ac | $6.9 \pm 3.7$ a |
| | ANOVA | | | | | | |
| | K | $p < 0.05$ | ns | $p < 0.001$ | $p < 0.001$ | ns | $p < 0.001$ |
| | Lines | ns | ns | $p < 0.05$ | ns | ns | Ns |
| | K*L | $p < 0.05$ | ns | ns | $p < 0.05$ | $p < 0.001$ | Ns |

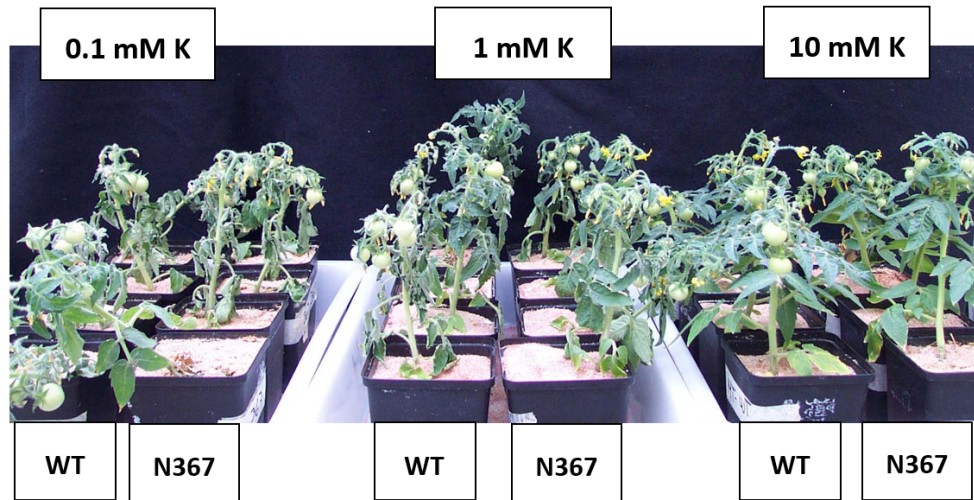

**Figure 1.** Plants of tomato lines WT and N367 growing with 0.1 (**left**), 1 (**center**) or 10 mM K (**right**) after withholding irrigation for two weeks.

The potassium supply also affected the ability for water acquisition by modifying hydraulic conductivity ($L_0$) in hydroponically grown plants. Root $L_0$ values showed significant differences among K treatments (Table 7). In low 0.1 mM K, root $L_0$ was greater than at higher K supplies. Greater $L_0$ values correlated with greater sap flows ($J_v$) but lower K contents in the xylem sap, indicating that plants absorbed more water at low-K supply to satisfy nutritional requirements. Significant differences between lines were also recorded at 0.1 mM K (Table 7). N367 plants had a higher K content in their xylem sap compared to controls when grown in sufficient K (1 and 10 mM). They also showed reduced sap flows ($J_v$) and hydraulic conductivity ($L_0$) at low 0.1 mM K.

Both water stress and K availability alter the concentration of amino acids in plants [19,20], some of which, like proline, have protective effects under drought stress [21]. Hence, we measured free amino acids in the leaves and flowers of water-stressed tomato plants submitted to two cycles of watering/drying under 0.1 and 10 mM K regimes. Indeed, nitrogen metabolism in MicroTom was also affected by K nutrition and water supply, as found in the variation in the free amino acids pool in leaves and flowers. At low K, the leaves from plants suffering water stress accumulated mostly arginine and proline (Supplementary Figure S2a) whereas at higher K supply, leaf proline content was significantly increased while arginine content remained rather unaffected. In flowers, glutamine, asparagine, proline and arginine were the main amino acids stored in both irrigated

and water-stressed plants and supplying more K increased the concentration of them all (Supplementary Figure S2b).

**Table 7.** Effect of available K on sap flow ($J_v$), root hydraulic conductivity ($L_0$) and xylem sap K concentration in hydroponically grown tomato lines WT and N367 at flowering stage. Means followed by the same letter are not significantly different (LSD Test, $p < 0.05$). K and L in ANOVA represent potassium and line factors, respectively.

| K (mM) | Line | $J_v$ | $L_0$ | Sap K |
|---|---|---|---|---|
| 0.1 | WT | 216.3 ± 24.6 a | 1221.9 ± 138.8 a | 3.5 ± 0.97 a |
| | N367 | 170.3 ± 19.2 b | 962.2 ± 108.2 b | 5.0 ± 1.1 a |
| 1 | WT | 60.3 ± 13.4 c | 370.2 ± 82.0 c | 26.7 ± 2.3 b |
| | N367 | 56.6 ± 18.0 c | 347.2 ± 110.6 c | 29.4 ± 6.4 bc |
| 10 | WT | 78.2 ± 19.6 c | 355.2 ± 88.9 c | 30.9 ± 2.1 c |
| | N367 | 82.2 ± 11.6 c | 373.6 ± 52.8 c | 35.7 ± 0.9 d |
| | ANOVA | | | |
| | K | $p < 0.001$ | $p < 0.001$ | $p < 0.001$ |
| | Lines | $p < 0.05$ | $p < 0.05$ | $p < 0.05$ |
| | K*Lines | $p < 0.05$ | $p < 0.05$ | Ns |

Units: $J_v$, sap flow, mg·g root fresh wt$^{-1}$ h$^{-1}$; $L_0$, root hydraulic conductivity, mg·g root fresh wt$^{-1}$ h$^{-1}$ MPa$^{-1}$; sap K, K concentration in the xylem sap, mM.

Priming plants with stress episodes improves plant adaptability to subsequent and repetitive stresses in a process known as acclimation, which enhances fitness and yield [15,16]. Hence, we tested how repetitive episodes of drying/watering (priming) affected the photosynthesis parameters, gas exchange and water-use efficiency of MicroTom plants at varying K availabilities. After reaching the flowering stage, plants growing with 0.1, 1 and 10 mM in the nutrient solution were submitted to either one cycle or four consecutive 2-week cycles of water withholding and rewatering (see Section 2). Plants experiencing only one water stress cycle were considered "nonacclimated", whereas primed plants that experienced three additional watering/drying cycles were deemed to be "acclimated". Physiological parameters were measured at the end of the one or four cycles of drying/watering and compared to the control plants that had been irrigated normally. The irrigation of control and treated plants was with complete nutrient solutions containing different K levels (0.1, 1 and 10 mM K). As expected, low water availability significantly reduced photosynthesis in acclimated and nonacclimated plants alike (Table 8). In plants suffering water stress, leaf photosynthetic and transpiration rates were limited by stomatal closure, and the increase in K supply did not prevent the drought-induced photosynthetic inhibition. However, increasing K availability to plants led to an improvement in intrinsic water-use efficiency (WUEi) in both irrigated and water-stressed plants (Table 8). Leaf K concentration was negatively associated with transpiration, stomatal conductance and photosynthesis (r = −0.54, r = −0.51, r = −0.46, $p < 0.001$, n = 36) but it was positively correlated with water-use efficiency (r = 0.50, $p < 0.001$, n = 36). The only positive association of leaf K contents was found in the photochemical reflectance index (PRI) (r = 0.61, $p < 0.001$, n = 44) at low K supply (see Supplementary Figure S3). At higher K supplies, leaf K contents were weakly and negatively associated with PRI (r = −0.27, $p < 0.05$, n = 80). When plants were grown with unlimited water supply, no association was found between K supply and stomatal conductance or transpiration rate (r = 0.06 and r = 0.09, n = 30). Priming plants led to an apparent improvement in the efficiency of C gain per mole of stomatal water lost, i.e., a greater increase in WUEi, in comparison with the irrigated controls (Table 8).

**Table 8.** Photosynthesis (*A*), transpiration rate (*E*), stomatal conductance ($g_s$) and intrinsic water-use efficiency ($WUE_i$) in plants that were watered every three days (irrigated), or received one cycle of drying/watering (nonacclimated) or four cycles (acclimated). Photosynthesis and gas exchange were measured after one or the four cycles of drying/watering, in comparison with plants that were irrigated normally. In the irrigation, plants had different K regimes (0.1, 1 and 10 mM K). Within each column, means (±standard deviation) followed by the same letter are not significantly different (LSD Test, $p < 0.05$). Units: *A*, µmol $CO_2$ m$^{-2}$ s$^{-1}$; *E*, mmol $H_2O$ m$^{-2}$ s$^{-1}$; $g_s$, mmol m$^{-2}$ s$^{-1}$; $WUE_i$, µmol $CO_2$·mmol$^{-1}$. K and W in ANOVA represent potassium and watering regime, respectively.

| K Supply (mM) | Water Treatment | *A* | *E* | $g_s$ | $WUE_i$ |
|---|---|---|---|---|---|
| | | Nonacclimated plants | | | |
| 0.1 | drought | 10.3 ± 0.8 a | 5.4 ± 0.1 a | 0.18 ± 0.01 a | 56.4 ± 7.5 a |
| | irrigated | 16.7 ± 0.8 b | 11.2 ± 0.7 b | 0.89 ± 0.09 b | 20.8 ± 6.0 b |
| 1 | drought | 7.5 ± 2.6 c | 2.4 ± 0.7 c | 0.09 ± 0.03 a | 87.4 ± 7.5 c |
| | irrigated | 15.2 ± 0.6 b | 8.0 ± 0.5 d | 0.40 ± 0.06 c | 40.5 ± 4.3 d |
| 10 | drought | 9.9 ± 2.5 a | 4.5 ± 1.1 a | 0.15 ± 0.05 a | 67.1 ± 8.5 a |
| | irrigated | 15.9 ± 1.9 b | 7.5 ± 1.4 d | 0.35 ± 0.10 c | 47.9 ± 10.7 ad |
| ANOVA | | | | | |
| K | | Ns | $p < 0.001$ | $p < 0.001$ | $p < 0.01$ |
| W | | $p < 0.001$ | $p < 0.001$ | $p < 0.001$ | $p < 0.01$ |
| K*W | | Ns | $p < 0.01$ | $p < 0.001$ | $p < 0.01$ |
| | | Acclimated plants | | | |
| 0.1 | drought | 5.1 ± 2.6 ac | 1.3 ± 0.6 a | 0.07 ± 0.04 a | 81.8 ± 40.6 a |
| | irrigated | 8.3 ± 1.3 b | 2.2 ± 0.5 b | 0.12 ± 0.05 b | 75.8 ± 21.3 a |
| 1 | drought | 3.4 ± 1.2 c | 0.6 ± 0.3 c | 0.02 ± 0.02 c | 214.9 ± 126.4 b |
| | irrigated | 5.5 ± 2.1 ac | 1.1 ± 0.5 a | 0.05 ± 0.02 ac | 128.9 ± 60.6 ad |
| 10 | drought | 5.8 ± 3.5 abc | 0.8 ± 0.3 ac | 0.03 ± 0.01 ac | 164.4 ± 52.3 b |
| | irrigated | 6.4 ± 1.9 ab | 1.0 ± 0.2 ac | 0.04 ± 0.01 ac | 148.3 ± 31.7 abd |
| ANOVA | | | | | |
| K | | $p < 0.05$ | $p < 0.001$ | $p < 0.001$ | $p < 0.01$ |
| W | | $p < 0.001$ | $p < 0.001$ | $p < 0.001$ | ns |
| K*W | | Ns | ns | ns | ns |

## 4. Discussion

In this work we have reassessed the contribution of K supply towards key physiological parameters in tomato under water stress with the distinctive advantage of comparing two isogenic lines differing in their ability to take up and compartmentalize K into vacuoles [13]. We found that high K in the nutrient solution improved the water contents in wild-type plants, particularly under limited watering conditions (Tables 5 and 6). A significant beneficial effect of K supplementation was also found in fruit number and yield in MicroTom plants (Tables 1 and 2). The yield of well-irrigated wild-type plants doubled with K supplement in the 1–10 mM range compared to 0.1 mM, and the increase was five-fold in water-stressed plants. However, the yield was most severely impaired by drought and this could not be overcome by K supplements since the yield of stressed plants remained five- to ten-fold lower than that of control plants at 1 and 10 mM K (Table 1). The benefit of K supplements was more intense in the transgenic line N367 under limited watering, although the yield of N367 plants approached that of the wild-type only at 10 mM, the highest concentration tested (Table 2). This differential behavior of N367 plants is likely related to the larger proportion of cellular K that is compartmentalized inside vacuoles in the transgenic line at the expense of the cytosolic pool [13], so that a higher K supplement is needed to off-set the enhanced compartmentation. In other words, greater K compartmentation into vacuoles may be detrimental when low K availability and water stress are combined.

The enhanced accumulation of K in vacuoles, achieved by the overexpression of tonoplast- and endosomal-localized NHX antiporters, has been shown to the increase the salt tolerance of tomato plants by improving K homeostasis and reducing salt-induced K

loss [13,24,25]. However, we show that the tomato line N367 with greater K accumulation failed to demonstrate a better performance under water stress compared to wild-type when plants suffered from water restrictions in spite of the consistently higher K contents in N367 plants (Tables 2 and 3). In the line N367, K sequestration in stems and roots (see Table 3) probably limited the nutrients available for assimilate redistribution to developing sinks (i.e., fruits) thereby reducing its K use efficiency (Table 4). However, the line N367 accumulated a significant amount of sugars in the leaves and stems at low K (Table 6). Sugars may accumulate either in response to water stress [26] or as a symptom of K deficiency [27,28]. The line N367 accumulates sugars under salt-stress or limited K supply [13,21], and yet it shows greater sensitivity to K deficiency in spite of similar water contents compared to wild-type MicroTom plants (Table 6). Sugar accumulation in N367, which results in lower assimilate transport to sinks, might explain the reduced fruit set and size and fruit yield (Table 2).

The limitation in leaf transpiration imposed by water scarcity led to a significant improvement in WUE at optimum K nutrition, mostly in acclimated plants, by reducing more water loss than photosynthetic carbon gain. Increasing K also significantly affected water economy by favoring stomatal closure (reducing water loss) under water stress and by reducing root hydraulic conductivity (with no water restrictions, see Table 7). However, greater K supply did not protect the photosynthetic rate under water stress as previously shown in wheat [3,4]. Only when the external K concentration was low, the leaf K contents were associated with an increase in PRI, i.e., leaf K might have a protective effect on the photosynthetic apparatus under drought [29].

High K improved carbon gain under water stress by increasing photosynthetic area, i.e., promoting shoot growth and therefore leaf expansion (Tables 1 and 3) and probably through the increased synthesis of protective osmolytes like proline (Supplementary Figure S2a). In our controlled conditions, K improved the shoot/root ratio (Table 1) and leaf WUE (Table 8) in water-stressed plants but it led to marginal increase in fruit yield (Table 2). The improvement in shoot growth and water-use efficiency by increasing K has also been found in other species (albeit by using different approaches [30–32]). The role of K in the osmotic potential and turgor generation required for expansive growth [1,33] may explain the significant increase in shoot growth of either water-deprived plants or irrigated plants when K supply was increased (Table 3).

The greater root hydraulic conductivity found at low K in nutrient solutions (Table 7) might be explained by K-deficiency sensing, which causes root ABA accumulation, increases in radial water flow and facilitates root water uptake [34–37]. Potassium deficiency also causes ethylene release and therefore ethylene-induced defective stomatal closure cannot be discarded [38,39] in the observed higher stomatal conductance and transpiration rate at low K (Table 8). As K deficiency progresses, root water conductivity diminishes by inhibition of aquaporins and K-uptake channels [12] and the accumulation of ABA in leaves may limit the effects of ethylene on stomata [40] leading to stomatal closure. On the other hand, increasing K supply may reduce root hydraulic conductance (Table 7), which might be responsible, at least partially, for the decrease in stomatal conductance (Table 8), as was found for other species [18].

The increased leaf proline contents under high K (Supplementary Figure S2a) may have provided both osmotic adjustment and radical scavenging to protect metabolic processes under water stress [22,41], leading to increased fruit setting and growth (Table 3). In flowers, the small changes in amino acid concentrations recorded either under water or K shortages (Supplementary Figure S2b) show them as preferential sinks and therefore less affected by stress [42].

A strong association between plant water content and K uptake and use is evident from the experiments shown here and in the related literature [11,12]. Undoubtedly, K contributes to maintaining plant water status through its effects on the regulation of root water conductivity and stomatal movements. It improves photosynthetic carbon assimilation under water scarcity, and thereby fruit yield, by improving shoot growth,

but had no effect on photosynthetic rates. However, the yield penalty caused by water limitation cannot be overcome by increasing the K supply. By using a tomato line (N367) able to accumulate higher K concentrations, the yield was not improved but in fact reduced by limiting K use efficiency (Table 4). The potassium stored in vegetative parts improved plant water contents but was a deterrent for getting better yields under drought as it was not available for the transport of assimilates into fruits.

When water and nutrient limitation affects different crops, the relationship between K availability and yield is not a simple one and the yield responses depend on several factors inherent to each species [43] and crop nutrient use efficiency [44]. In tomato, K may affect water acquisition and economy as well as C gain, and its partitioning in a close interplay and nutrient use efficiency is of paramount importance for fruit yield. A greater K accumulation (N367) was not translated into an improvement of plant growth or fruit yield when plants suffered water stress, but indeed proved to be a disadvantage by reducing plant growth and consequently fruit setting and size.

In conclusion, K supplementation had a protective effect on the yield of tomato plants under limited watering, but this benefit was insufficient to off-set the yield loss induced by water shortage. Increasing the plant K contents through enhanced compartmentation into vacuoles failed to protect yield under drought stress, even though it could improve some physiological traits.

**Supplementary Materials:** The following is available online at https://www.mdpi.com/2311-7524/7/2/20/s1. Figure S1: Scheme showing the different regimes of drying/rewatering used to measure the effect of K availability on the yield and acclimation of water-stressed plants. Figure S2: Free amino acids in (a) leaves and (b) flowers from tomato plants with or without water stress when grown at low (0.1 mM K) or high K (10 mM K). Figure S3: Effect of K and watering treatments on (a) leaf K concentration and (b) photochemical reflectance index in two tomato lines (WT and N367).

**Author Contributions:** E.O.L. designed and supervised the research; A.D.L., M.C. (Mireia Corell), M.C. (Mathilde Chivet) performed experiments and analyses; all authors analyzed and discussed the data; E.O.L. and J.M.P. wrote the manuscript and obtained the funding. All authors have read and agreed to the published version of the manuscript.

**Funding:** This work was supported by grant RTI2018-094027-B-I00 from the Spanish Agencia Estatal de Investigación (AEI), of Ministerio de Ciencia, Innovación y Universidades (MCIU), and cofinanced by the European Regional Development Fund, to E.O.L. and J.M.P.

**Institutional Review Board Statement:** Not applicable.

**Informed Consent Statement:** Not applicable.

**Data Availability Statement:** Data supporting the present results are available on request to the corresponding author.

**Acknowledgments:** The authors thank R. Fernández, A. Cifarelli and D.M. Cabrera for their contributions to different experiments while in periods of training, and Francisco J. Quintero and Anna M. Lindahl for critical reading of the manuscript. F.J. Quintero produced the scheme in Supplementary Figure S1.

**Conflicts of Interest:** The authors declare no conflict of interest.

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
