# Peer review of "Reassessing the Role of Potassium in Tomato Grown with Water Shortages"

_horticulturae, doi:10.3390/horticulturae7020020_

Round 1
Reviewer 1 Report
Thank you for giving me the opportunity to read and review the manuscript “Re-assessing the role of potassium in tomato grown with water shortages.” by Luca et al.
The research topics and questions are interesting me as a reader of horticulturae.
However, the current manuscript is difficult to review due to insufficient explanation of methods and results. So the current manuscript must be improved before peer review. The details are shown below. I hope you find at least some of my comments useful -Thanks.
Material and methods: I was confused as to what was different between Experiments 1.2.3. First of all, it is necessary to organize and explain the experimental design.
L77-78: I did not understand the water stress treatments that the authors did. How many days did authors irrigate in one "watering / drying cycle"? Is "during 2 weeks followed by three watering/drying cycles” mean that the authors irrigate every two days?
L81-82: Which experiment and which treatments was taken as "After a single water-stress cycle plants were considered ‘non-acclimated’ whereas ‘acclimated’ plants experienced 4 watering/drying cycles"? The sentence just before says that plant of water stress treatment was taken three watering/drying cycles.
How did the authors determine that the plant was "adapted"?
L84-114: When were the measurements of methods 2.2 leaf water status - 2.5 K concentration and 2.7 free amino acid contents conducted during the experimental period, respectively?
L95: What is the unit of root hydraulic conductivity (L0)?
L109: What is the contribution of measurements of free amino acid in flower and leaf to this study? Please explain it in the Introduction or here.
L115-118: It is necessary to explain what kind of statistical analysis was performed using which result.
Results: I cannot know which result each experiment 1.2.3 is associated with.
Table 1: The caption description is incorrect. I don't think this table includes the results of the shoot K concentration.
L153-164: The explanation uses four types of words: "K supply", "K available", "K concentration", and "K content", but it is difficult to tell whether it indicates K treatment or K concentration. Furthermore, although only the relationship between leaf K concentration and yield is mentioned, is it necessary to mention the relationship between stem and root K concentration?
L165-169: I cannot find the results showing that the stem water content did not change between the K treatment.
L190-193, Table 8: There was no explanation on how to measure sap flow and SapK in the Material and methods part.
Due to uncertainties about the results so far, the discussion should be refrained from peer review.
Author Response
"Please see the attachment."

Reviewer 2 Report
Dear Editors,
I would like to thank you very much for the invitation as a reviewer for the manuscript horticulturae-992848 “Re-assessing the role of potassium in tomato grown with water shortages.” The article is interesting and dedicated to the vital problem of potassium nutrition of plants. The authors did a good study, which should be highly appreciated.
The article meets the requirements of the Horticulture journal, but it needs some explanations and clarifications (minor changes):
- The scientific hypothesis, as well as the aim of the study, should be more clearly stated.
- The authors mentioned in the 2. Materials and Methods section (subsection 2.1.) about Experiments 1, 2, and 3 (lines 66 and 67) but in subsection 2.8. Statistical analysis only Experiment 1 and 2 (lines 117-118).
- Subsection 2.3. is mentioned photochemical reflectance index measurement. There are no data in tables about the values of PRI. The Result section is given only correlation coefficients of PRI with leaf potassium in plants with low K supply.
- The present study does not address salt tolerance, in section 4. Discussion, the authors discuss the problem of salt stress based only on other studies’ results.
- The authors wrote about the positive role of potassium, and its impact on yield and other studied parameters (262-271) didn’t mark the rate of the nutrient needed for such a result.
- It would be better if the authors will add the section Conclusions to emphasize this study results
- Journal references must cite the digital object identifier (DOI) where available. In the list of references, only 4 references are with DOI.
The reference No. 16, there are no page numbers
Author Response
"Please see the attachment."

Reviewer 3 Report
This paper presents a study on the role of potassium in tomato grown with water shortages. The results demonstrate that K could improve some plant physiological characteristics although the plant not is able to overcome yield restrictions imposed by water stress. Although the paper tackles an interesting topic, the study shows several flaws that hinder the publication of the manuscript in the current form. The main flaws are related to the material and methods, which is not enough detailed (e.g. hydroponic experiment is not described, nor the duration of watering/drying cycles, nor the aeration conditions…), and to the results section, which is difficult to follow due to the lack of connexion with the above-mentioned section (i.e. the different experiments are not clearly represented in the tables of results).
Author Response
"Please see the attachment."

Round 2
Reviewer 1 Report
The revised manuscript is now easier to follow for a reader. However, the authors will need to evaluate the results more objectively in their discussions and conclusions. The details are shown below. I hope you find at least some of my comments useful -Thanks.
Q1.9. L115-118: It is necessary to explain what kind of statistical analysis was performed using which result.
R1.9. You were right. The mistake has been corrected.
The explanation in the statistical analysis section is still insufficient. For example, the results in Table 1 seem to have used the LSD test and ANOVA, but this section generally describes what statistical method was used to test the effect on all the results. On the other hand, the text added to L176-180 is unnecessary because it overlaps with the method.
Q1.10. Results: I cannot know which result each experiment 1.2.3 is associated with.
R1.10. We explain now the experimental setup of each experiment before the data set is presented. We hope this is easier to follow.
The material and method (M&M) descriptions have been organized so that there is no need to duplicate explanations in the result. It is redundant and can be deleted or added into M&M. (L217-220, L264-266, l281-291). The purpose and method of the experiment should be well explained in the method section. In the current method, there is a part where the explanation of "what you want to test" is not enough.
new comments be after here:
L330-331: Do these results accompany Table 8?
L363-365: According to LSD test, in Tables 5 and 6, the plant water content is 0.1 <1 = 10 in the K supply plot, and similarly in Tables 1 and 2, the yield is basically 0.1 <1 = 10 in the K supply plot. The discussion here will need to be reconsidered.
L387-388: In Table 2, there is a 30-fold increase in yield when K supply = 1 and a 45-fold increase when K supply = 10, compared to when K supply = 0.1, and a significant effect has been verified in the LSD test. It seems to mean that there is a proper amount of K supply, not that it has not improved. The following text of L389-390 will also need to be reconsidered. Please discuss the results objectively.
L395-396: I do not understand the mean that "in spite of similar K tissue contents than the WT".
The conclusion should also be reviewed.
Author Response
Q1.1. The revised manuscript is now easier to follow for a reader. However, the authors will need to evaluate the results more objectively in their discussions and conclusions. The details are shown below. I hope you find at least some of my comments useful -Thanks.
Sure, the comments have been very useful, and we appreciate the contribution and guidance of Reviewer 1 to improve the manuscript. We have revised the Discussion and Conclusions as suggested.
Q1.2. The explanation in the statistical analysis section is still insufficient. For example, the results in Table 1 seem to have used the LSD test and ANOVA, but this section generally describes what statistical method was used to test the effect on all the results. On the other hand, the text added to L176-180 is unnecessary because it overlaps with the method.
We are not sure what the problem and the recommendation is here because we stated the statistical analyses applied in every Table. We have nevertheless added a sentence to section ‘Statistical analysis’ in M&M indicating that ‘Mean comparisons were made with the Least Significant Difference (LSD) test when the F-values from the Analysis of Variance (ANOVA) were statistically significant’.
Last sentence in this comment relates to the next question Q1.3.
Q1.3. The material and method (M&M) descriptions have been organized so that there is no need to duplicate explanations in the result. It is redundant and can be deleted or added into M&M. (L217-220, L264-266, l281-291). The purpose and method of the experiment should be well explained in the method section. In the current method, there is a part where the explanation of "what you want to test" is not enough.
A consistent complaint of Reviewers regarding the original manuscript was the difficulty to match the different experiments described in M&M with the specific data sets presented in Results as Tables. In the revised manuscript we tried to correct this problem by first giving a general description of the experimental approach and procedures under the heading ‘Stress treatments’ in M&M, and then a concise description of the specific experiment before each data set was presented in Results so that the readers should not be confused regarding which experiment applies to each data set. It seems that, at least for Reviewer 1, we have achieved this goal. Moreover, in this way, the reader does not need to go back to M&M the recapitulate the experimental setup to understand the data presented in the Tables. We have done this without a major increase in the manuscript length, which is a good trade-off. We are afraid that if we reverted back to physically separate the explanation of the different experiments performed from the data sets and Tables, we may again compromise the readability and clarity of the manuscript.
We also argue that authors should have certain freedom for deciding how to organize the manuscript as long as it is comprehensible for the general readership. Hence, we request to keep the information flow as it is the revised manuscript in spite of certain redundancy between M&M and Results.
Q1.4. L330-331: Do these results accompany Table 8?
Yes, they do. The sentence was separated from the rest of the paragraph after inserting Table 8. This has been corrected and further clarified by indicating that the sentence pertains to Table 8 (L290-292).
Q1.5. L363-365: According to LSD test, in Tables 5 and 6, the plant water content is 0.1 <1 = 10 in the K supply plot, and similarly in Tables 1 and 2, the yield is basically 0.1 <1 = 10 in the K supply plot. The discussion here will need to be reconsidered.
As suggested, we have revised the Discussion in this part to reflect more precisely the results (L306-320).
Q1.6. L387-388: In Table 2, there is a 30-fold increase in yield when K supply = 1 and a 45-fold increase when K supply = 10, compared to when K supply = 0.1, and a significant effect has been verified in the LSD test. It seems to mean that there is a proper amount of K supply, not that it has not improved. The following text of L389-390 will also need to be reconsidered. Please discuss the results objectively.
The words ‘compared to the wild-type’ were missing in the conflicting sentence (which has been deleted now). As suggested, we have reorganized the Discussion and merged our interpretation of the results pertaining to the comparative yield of wild-type and N367 plants in two consecutive paragraphs (L306-334).
Q1.7 L395-396: I do not understand the mean that "in spite of similar K tissue contents than the WT".
Sorry, that was a typo. We meant similar water contents. K contents are consistently greater in N367 than in control Microtom plants (references 13 and 21), although differences may not be always statistically significant (see Table 3).
Q1.8. The conclusion should also be reviewed.
We have revised and extended our conclusions to make them clearer. The Abstract has also been revised to make it shorter (as per the journal style) and with clearer conclusions (384-387).
Reviewer 3 Report
Although the manuscript has considerably improved respect to the original, the study continues showing several flaws related to the material and methods, which is not enough detailed (e.g., hydroponic experiment is not described, nor the quantification of drought conditions, nor the number of consecutive cycles of drying and re-watering of the different experiments…), which does that the results are difficult to follow due to the lack of connexion with the above-mentioned section, even repeating the experimental conditions at the beginning of the results section (which should be deleted). Maybe, adding a figure (diagram) which explains the sequence of the events could clarify the outline of the experiments.
Author Response
Although the manuscript has considerably improved respect to the original, the study continues showing several flaws related to the material and methods, which is not enough detailed (e.g., hydroponic experiment is not described, nor the quantification of drought conditions, nor the number of consecutive cycles of drying and re-watering of the different experiments…), which does that the results are difficult to follow due to the lack of connexion with the above-mentioned section, even repeating the experimental conditions at the beginning of the results section (which should be deleted). Maybe, adding a figure (diagram) which explains the sequence of the events could clarify the outline of the experiments.
We do not agree with these negative remarks. In the revised manuscript (Revision 1), we improved the description of the different experiments by first giving a general description of the experimental approach and procedures under the heading ‘Stress treatments’ in M&M, and then a concise description of the specific experiment before each data set is presented in Results. Moreover, the number of cycles of drying and re-watering was specifically stated in each Table legend. Please check that in Table 1 we wrote ‘Control plants (irrigated) were compared to plants submitted to six cycles of drying/watering after flower initiation (drought)’; in Table 2 ‘in two lines (WT and N367) differing in K uptake after six cycles of limited watering’; in Table 3 ‘two tomato lines (WT and N367) at harvest after receiving six cycles of limited watering’; in Table 4 ‘two tomato lines (WT and N367) grown with different K availabilities and subjected to six cyclical droughts’; Table 5 ‘Control plants (irrigated) were compared to plants submitted to six cycles of drying/watering after flower initiation (drought)’; Table 6 ‘WT and N367 lines at harvest when grown with different K concentrations and subjected to six cyclical drought periods’, and then in Table 8 ‘in plants that were watered every three days (irrigated), or received one cycle of drying/watering (non-acclimated) or four cycles (acclimated)’.
The experiment in hydroponics was described in the subsection ‘Root hydraulic conductivity’ of M&M and in the legend to Table 7 where data was presented. We have nevertheless made this even more clear in the newly revised manuscript, where we wrote: ‘Root hydraulic conductivity (Lo) was estimated by measuring root exudation rates in de-topped plants [18] grown hydroponically with nutrient solutions containing different K concentrations [13]’ (L125-126).
Round 3
Reviewer 1 Report
The revised manuscript has been improved in line with my previous comments in the discussions and conclusions. However, as other reviewers also continue to point out, M & M and Results are still not presented concisely and precisely. The details are as follows.
Q1.3. The material and method (M&M) descriptions have been organized so that there is no need to duplicate explanations in the result. It is redundant and can be deleted or added into M&M. (L217-220, L264-266, l281-291). The purpose and method of the experiment should be well explained in the method section. In the current method, there is a part where the explanation of "what you want to test" is not enough.
Author's reply: A consistent complaint of Reviewers regarding the original manuscript was the difficulty to match the different experiments described in M&M with the specific data sets presented in Results as Tables. In the revised manuscript we tried to correct this problem by first giving a general description of the experimental approach and procedures under the heading ‘Stress treatments’ in M&M, and then a concise description of the specific experiment before each data set was presented in Results so that the readers should not be confused regarding which experiment applies to each data set. It seems that, at least for Reviewer 1, we have achieved this goal. Moreover, in this way, the reader does not need to go back to M&M the recapitulate the experimental setup to understand the data presented in the Tables. We have done this without a major increase in the manuscript length, which is a good trade-off. We are afraid that if we reverted back to physically separate the explanation of the different experiments performed from the data sets and Tables, we may again compromise the readability and clarity of the manuscript.
We also argue that authors should have certain freedom for deciding how to organize the manuscript as long as it is comprehensible for the general readership. Hence, we request to keep the information flow as it is the revised manuscript in spite of certain redundancy between M&M and Results.
My new comments: I disagree with the authors' claims. This is because the basic premise of what should be written in each section of the research paper is ignored. M&M section should be an explanation and clear organization of experimental procedures and dataset handling should be provided, and Results section should be a concise and accurate description of the experimental results, their interpretation, and the experimental conclusions that can be drawn. This is not my personal claim, but is clearly stated in 'Instructions for Authors' of this journal.
In M ​​& M, the authors need to organize the experimental design and explain the reproducible method. Redundant explanations that ignore what should be written in each section only tire the readers, and it can be said that the authors have abandoned their responsibility to properly compose the manuscript.
It would be helpful for the reader to add an item to organize and summarize the experimental objectives and dataset descriptions currently displayed before each result, or to have a diagram showing the scheme of those study designs.
Author Response
Reviewer 1
Q1. The revised manuscript has been improved in line with my previous comments in the discussions and conclusions. However, as other reviewers also continue to point out, M & M and Results are still not presented concisely and precisely. The details are as follows.
I disagree with the authors' claims. This is because the basic premise of what should be written in each section of the research paper is ignored. M&M section should be an explanation and clear organization of experimental procedures and dataset handling should be provided, and Results section should be a concise and accurate description of the experimental results, their interpretation, and the experimental conclusions that can be drawn. This is not my personal claim, but is clearly stated in 'Instructions for Authors' of this journal.
In M & M, the authors need to organize the experimental design and explain the reproducible method. Redundant explanations that ignore what should be written in each section only tire the readers, and it can be said that the authors have abandoned their responsibility to properly compose the manuscript.
It would be helpful for the reader to add an item to organize and summarize the experimental objectives and dataset descriptions currently displayed before each result, or to have a diagram showing the scheme of those study designs.
R1. We have revised again the description of M&M with the help of two colleagues, not related to this work, who have read the Ms critically. They find that the newly revised M&M we are submitting now are clear enough to warrant reproducibility. They also concur with our opinion that adding a short description of the purpose and design of each experiment before the corresponding data set are presented in Results are helpful to the reader and does not make the manuscript unnecessarily verbose. Finally, we have added as Supplementary Figure 1 a diagram of the design and execution of the main experiments of this work, aiming, respectively, to assess the effect of K supplements in the yield of tomato plants submitted to recurrent water stress treatments, or the ability of K supplements to improve the acclimation of tomato plants to water stress. We hope that these improvements and new figure are satisfactory.

Reviewer 3 Report
The material and methods is not enough detailed yet, which makes difficult that the results can be reproducible.
Author Response
Q1. The material and methods is not enough detailed yet, which makes difficult that the results can be reproducible.
R1. We have revised again the description of M&M with the help of two colleagues, not related to this work, who have read the Ms critically. They find that the newly revised M&M we are submitting now are clear enough to warrant reproducibility. Moreover, and in line with a previous request, we have included as Supplementary Figure 1 a diagram of the design and execution of the main experiments of this work, aiming, respectively, to assess the effect of K supplements in the yield of tomato plants submitted to recurrent water stress treatments, or the ability of K supplements to improve the acclimation of tomato plants to water stress. We hope that these improvements and new figure are satisfactory.
Round 4
Reviewer 3 Report
I appreciate authors' efforts for improving the manuscript. Now it the methodology is clearer and the manuscript more argued.